# Liver Acinus Dynamic Chip for Assessment of Drug-Induced Zonal Hepatotoxicity

**DOI:** 10.3390/bios12070445

**Published:** 2022-06-23

**Authors:** Dohyung Kwon, Geonho Choi, Song-A Park, Sungwoo Cho, Sihun Cho, Sungho Ko

**Affiliations:** 1Department of Biotechnology, CHA University, 335 Pangyo-ro, Bundang-gu, Seongnam-si 13488, Korea; 24dohyung@chauniv.ac.kr (D.K.); geonhochoi@chauniv.ac.kr (G.C.); sapark@chauniv.ac.kr (S.-A.P.); csw2669@chauniv.ac.kr (S.C.); imonariga@chauniv.ac.kr (S.C.); 2Humanase Co., Ltd., 335 Pangyo-ro, Bundang-gu, Seongnam-si 13488, Korea

**Keywords:** organ-on-a-chip, hepatic zonation, liver acinus, hepatotoxicity, acetaminophen

## Abstract

Zonation along the liver acinus is considered a key feature of liver physiology. Here, we developed a liver acinus dynamic (LADY) chip that recapitulates a key functional structure of the liver acinus and hepatic zonation. Corresponding to the blood flow from portal triads to the central vein in vivo, gradual flow of oxygen and glucose–carrying culture medium into the HepG2 cell chamber of the LADY chip generated zonal protein expression patterns in periportal (PP) zone 1 and perivenous (PV) zone 3. Higher levels of albumin secretion and urea production were obtained in a HepG2/HUVECs co-culture LADY chip than in HepG2 mono-culture one. Zonal expression of PEPCK as a PP marker and CYP2E1 as a PV marker was successfully generated. Cell death rate of the PV cells was higher than that of the PP cells since zonal factors responsible for metabolic activation of acetaminophen (APAP) were highly expressed in the PV region. We also found the co-culture enhanced metabolic capacity to process APAP, thus improving resistance to APAP toxicity, in comparison with HepG2 mono-culture. These results indicate that our LADY chip successfully represents liver zonation and could be useful in drug development studies as a drug-induced zonal hepatotoxicity testing platform.

## 1. Introduction

The U.S. National Institutes of Health (NIH) in 2017 reported that 95% of drugs that have successfully passed in animal testing failed in clinical trials [1]. The results obtained from animal models cannot be fully translated to humans due to different pharmacokinetics, pharmacodynamics, and interspecies genetic variations [2,3,4]. In particular, many drug candidates approved in animal testing have shown hepatotoxicity in clinical trials due to the significant differences of drug metabolism between animal and human livers [5,6]. Among 462 medicinal products withdrawn from the pharmaceutical market between 1953 and 2013, hepatotoxicity (81 cases; 18%) was the most common reason [7]. In addition, traditional in vitro cell cultures cannot adequately mimic the physiological microenvironment of cells in vivo [8]. Furthermore, the lack of cell-to-cell and cell-to-matrix interactions in the in vitro cultures often causes the loss of cell function, sometimes resulting in non-predictive data for in vivo responses [8,9,10]. Therefore, the importance of accurate prediction of drug-induced liver injury in preclinical stages has been emphasized, which increases the need for human liver physiology-relevant in vitro models as alternatives to the conventional models such as animal testing and in vitro culture [11,12].

Organ-on-a-chip is a promising interdisciplinary technology that allows mimicking in vivo physiology; therefore, it can be used as an alternative to animal models and in vitro culture for efficacy and toxicity testing in the field of drug discovery, bridging the gap between preclinical and clinical testing [13,14]. Several recent studies have developed liver-on-chips that mimic the physiological structure of the liver to improve hepatic functions in comparison with a 2D monolayer culture and to screen for drug-induced liver injury. Some of them have focused on mimicking liver lobule structure, which is a building block of liver tissue and consists of a portal triad, hepatocytes arranged in linear cords between a capillary network and a central vein [15], by a field-induced dielectrophoresis [16], the digital light process-based bioprinting [17], and pneumatic microvalve system [18].

The functional and metabolic gradients in the liver occur from the periportal (PP) region (zone 1) of the lobule sinusoid to the perivenous (PV) region (zone 3), which is known as “liver zonation” [19,20,21]. Generally, hepatocytes (parenchymal cells of the liver) in zone 1 are efficient at oxidative metabolism, fatty acid oxidation, gluconeogenesis, bile acid extraction, ammonia detoxification, and urea and glutathione (GSH) conjugation, whereas zone 3 hepatocytes are efficient at glycolysis, liponeogenesis, and CYP450-mediated metabolism of xenobiotics [22,23,24,25]. Since the mimicking of liver zonation is crucial for reliable drug toxicity testing, some studies have developed microfluidic models recapitulating liver zonation to screen for drug-induced zonal hepatotoxicity, such as a liver acinus microphysiology system that demonstrates drug-induced hepatotoxicity in different zones [26], and a chip that is capable of dynamically creating metabolic patterns across the length of a microchamber of liver tissue [25]. A liver acinus (a functional unit of the liver lobule) is typically diamond-shaped with two triangular sections of the adjacent lobules, including two central veins at the opposite corners and two portal triads crossing the center and is divided into three zones in the direction of blood flow from portal triads to the central vein [27,28]. A co-culture system of hepatocytes with non-parenchymal cells (sinusoidal endothelial cells, Kupffer cells, stellate cells, and lymphocytes) can enhance relevance to the heterogeneity of the liver in vivo [29,30]. However, the above-mentioned microfluidic zonation models did not simultaneously materialize the diamond-shaped microstructure of the liver acinus and a co-culture system of hepatocytes with non-parenchymal cells.

In this work, we present a liver acinus dynamic (LADY) chip that recapitulates the half of the diamond shape of the liver acinus to induce hepatic zonation and allows for 3-dimensional (3D) co-culture of hepatocytes with endothelial cells to accurately test drug-induced zonal hepatotoxicity. The metabolic zonation was successfully formed in our LADY chip by a medium flow that corresponds to blood flow from portal triads to the central vein in vivo. This zonation induced relatively high expression of PP marker, phosphoenolpyruvate carboxykinase (PEPCK), in zone 1 hepatocytes and PV marker, cytochrome P450 2E1 (CYP2E1), in zone 3 hepatocytes, which caused a gradient of drug-induced hepatotoxicity across the length of the culture chamber. Co-culture of hepatocytes with endothelial cells enhanced resistance to drug cytotoxicity in comparison with hepatocyte mono-culture. These results demonstrate that our LADY chip has successfully recapitulated the microenvironment of the liver acinus, thus can be a useful in vitro platform for drug screening and toxicity testing. The novelty of the LADY chip is the first model to recapitulate liver zonation in microphysiological structure of the liver acinus. Similar to sinusoidal endothelial cells, the unique weir structure in the chip could effectively separate HepG2 and HUVEC cells for their co-culture, supply fresh culture medium, and reduce shear stress to HepG2 cells. Figure 1 shows our strategy to construct the LADY chip.

## 2. Materials and Methods

### 2.1. Design and Fabrication of the Microfluidic Device

The microfluidic LADY chip was designed and fabricated to mimic the 3D structure of the liver acinus to form hepatic zonation. The width of the entrance of the hepatocyte chamber is 80 µm, and the end of it is 90 µm; this design reflects the microenvironment of the hepatic cord, which is composed of 2 or 3 rows of hepatocytes (each 20–30 µm in diameter) along a liver sinusoid [31]. The length of the hepatocyte chamber is 1.5 mm reference from the actual size of the normal human liver lobule, 1 to 2 mm in diameter from a central vein to the other one of the adjacent lobule [32]. The hepatocyte chamber and the sinusoidal chamber are separated by a weir structure 30 µm thick, with a 7 µm gap above the bottom. This structure was designed to block cells from flowing out of each chamber. The overall size of the LADY chip is 30 mm × 14 mm, and the height of the inner microchamber is 67 µm. The PP well, which mimics portal triads, acts as a medium inlet, and the PV well, which mimics the central vein, acts as a cell inlet and a medium outlet, receiving medium flow from the PP well. This design simulates flow from portal triads to the central vein, which recapitulates the radial flow pattern in the liver lobule.

The LADY chip consists of a microchannel-patterned bottom layer and a top layer for the media reservoir, which enables gravity-driven medium flow (Figure 1). The two layers were fabricated from polydimethylsiloxane (PDMS, Sylgard 184, Dow Corning^®^, Midland, MI, USA), which was replicated by using a SU-8 micromold processed by photolithography. PDMS was prepared by mixing the elastomer base and curing agent from a Sylgard 184 silicone elastomer kit at a 10:1 ratio by weight. The mixture was poured onto the micro-patterned mold to fabricate micro-patterned bottom layers and on a bare petri dish to fabricate top medium reservoir layers. They were cured at 80 °C on a hot plate for 12 h, and an adhesive polymer mixture was peeled off from the mold and the dish. The bottom layers were cut into pieces of appropriate size, and an inlet and an outlet were punched using a 3 mm puncher. The top layers were cut to the same size as the bottom layers, and the PP and PV wells for the media reservoir were punched using an 8 mm puncher at the same positions as the inlet and the outlet of the bottom layer, respectively. Next, bottom layers, top layers, and glass slides were cleaned with isopropyl alcohol and sterilized under ultra violet (UV). The bottom layers and glass slides were then treated with oxygen plasma using a CUTE plasma treatment system, and each bottom layer was placed on a glass slide so that the surface with the microchannel contacted the slide. Then, the slide-bonded bottom layer and a top layer were treated with oxygen plasma, and the top layer was placed on the bottom layer so that each reservoir well was precisely aligned between the layers. Finally, both the PP well and the PV well were filled with DPBS without divalent cations, such as calcium and magnesium to maintain the hydrophilicity of the microfluidic chamber surface so that the cells could be easily injected into the cell chamber. Any aqueous buffer available for cell culture can be used to keep the hydrophilicity. DBPS was replaced with cell culture medium before cell injection into the chamber.

### 2.2. Cell Culture and Media

Human liver cell line HepG2 was purchased from the Korean Cell Line Bank (KCLB, Seoul, Korea) and cultured with MEM (Minimum Essential Medium, Eagle, high-glucose, Gibco) supplemented with 25 mM HEPES (4-(2-hydroxyethyl)-1-piperazineethanesulfonic acid, Gibco), 25 mM NaHCO_3_ (sodium bicarbonate, Sigma-Aldrich, St. Louis, MO, USA), 10% (*v*/*v*) fetal bovine serum (FBS, Gibco), and P/S (100 U ml^−1^ penicillin/100 U ml^−1^ streptomycin, Sigma-Aldrich). HUVECs were obtained from ATCC^®^ (EA.hy926, CRL-2922, Manassas, VA, USA) and maintained in Dulbecco’s Modified Eagle’s Medium (DMEM, ATCC^®^ 30-2002™) supplemented with 10% FBS. HUVECs are the most commonly used models as endothelial cells as they are known to have enough endothelium function, such as cell type homogeneity, well-characterized surface markers, late endothelial progenitor cell characters, and obtainable huge cell numbers via rapid expansion property [33]. The weir structure of our device could act as fenestrated sinusoidal endothelial cells by supplying fresh culture medium and reducing shear stress to cells.

HepG2 cells and HUVECs were used at a passage of 3–6 and 50–55, respectively. They were cultured in an atmosphere of 5% CO_2_ in air in an incubator at 37 °C and usually subcultured at a ratio of 1:4 every two days. When the cells reached 80% confluence or right before being used for the LADY chip, they were harvested by 0.25% trypsin–EDTA treatment and centrifugation at 1000 rpm for 3 min.

### 2.3. Formation of the Liver Zonation on a Chip

To form metabolic zonation in the LADY chip, cells were stacked in each cell chamber in 3D to control the flow rate and produce gradients of oxygen and glucose from zone 1 to 3 by forming a gradual flow of culture medium from the PP well to the PV well. To arrange the two types of cells in a liver acinus–mimetic pattern, first, HUVECs were injected into the PP well (Figure 1c). For this purpose, 1 × 10^4^ HUVECs were transferred into a 1.5 mL epi-tube and then pelleted in mini-centrifuge. The supernatant was carefully removed without disturbing the cell pellet, and the pellet was resuspended in 5 μL of HUVEC culture medium. The resuspended cells were injected into the PP well using a pipette. After cell flow was stopped by the weir structure, which acted as a fenestrated barrier between hepatocytes and the hepatic sinusoid, the device was immediately placed in a 5% CO_2_ incubator at 37 °C for 1.5 h to ensure the stable attachment of HUVECs in the sinusoidal chamber.

Afterwards, HepG2 cells were injected into the hepatocyte chamber (Figure 1c). For this purpose, 1 × 10^5^ HepG2 cells were transferred into a 1.5 mL epi-tube and then pelleted in a mini-centrifuge. The supernatant was carefully removed without disturbing the cell pellet, and the pellet was resuspended in 2 μL of HepG2 cell culture medium. The resuspended cells were injected into the PV well using a pipette. As soon as the flow of cells was stopped by the weir structure, the device was immediately placed in a 5% CO_2_ incubator at 37 °C to stabilize the cells in the hepatocyte chamber. One hour later, the PP and PV wells were filled with 50 μL of HepG2 cell culture medium by adding 10 μL five times into each well in turn, and the device was incubated for an additional 4 h to fully stabilize the cells. Then, 200 μL HepG2 cell culture medium was added into each well overnight, and the medium was changed to 300 μL (PP well) and 150 μL fresh medium (PV well) on a daily basis from culture day 1.

A previous study reported that liver is composed of parenchymal cells (80%) and non-parenchymal cells (NPC) (20%), and liver sinusoidal endothelial cells (LSECs) account for 50% among NPC population [34]. Thus, the ratio of hepatocytes to LSECs can be calculated as about 8:1. Based on this ratio, we used a 10:1 ratio of HepG2 cells to HUVECs in this study with taking error into account.

### 2.4. Liver Acinus–Mimetic Cell Pattern in LADY Chip

To observe the liver acinus–mimetic pattern and the maintenance of cell co-culture in the microfluidic device, we performed lentiviral transduction of the green fluorescent protein (GFP) gene into HepG2 cells and the red fluorescent protein (RFP) gene into HUVECs. To transduce the GFP gene, a lentiviral vector was generated by co-transfection of 293FT cells with three target plasmids: a transfer vector (pSin-EF2-eGFP), an envelope vector (pHDMG), and packaging vectors (pHDMG-Hgpm2, pRC/CMV-Rev1b, pHDM-tat1b). We used Lipofectamine 3000 reagent diluted in Opti-MEM™ I medium (both from ThermoFisher Scientific, Waltham, MA, USA) to produce high-quality lentivirus. After incubated plasmids were treated in 293FT cells, supernatant medium containing lentivirus was harvested every 24 h for 4 days. Following removal of debris by filtering through a 45 μL pore size filter, lentivirus was collected by ultracentrifugation (Beckman Coulter ultracentrifuge) at 20,000 rpm for 2 h. Serial dilutions of lentivirus were used to infect HepG2 cells, and culture medium was changed daily for 2 more days. GFP-positive fluorescent HepG2 cells were identified by flow cytometry using a MoFlo XDP High speed Cell Sorter (Beckman Coulter).

To transduce the RFP gene, a lentiviral vector was generated by co-transfection of 293FT cells with a transfer vector (pSin-EF2-mcherry) and the envelope and packaging vectors described above. The following experimental procedure was the same as above.

GFP-HepG2 cells and RFP-HUVECs were injected into the LADY chip, and confocal images were taken by an ImageXpress Micro Confocal High-Content Imaging System (IXM-C, Molecular Devices, San Jose, CA, USA).

### 2.5. Albumin and Urea Assay

To determine the levels of major liver-specific function markers, albumin synthesis and urea secretion were measured by ELISA and compared between the HepG2 mono-culture model and HepG2/HUVECs co-culture model. The culture supernatant in the PV well was harvested daily from day 1 and stored at −80 °C before analysis. Assays were performed according to the manufacturers’ instructions by using a human albumin ELISA kit (Bethyl Laboratories, Montgomery, TX, USA) and a Urea Assay Kit (Abcam, ab83362). The absorbance was measured by an ELISA microplate reader (Synergy H1, BioTeK, Winooski, VT, USA) at a wavelength of 450 nm.

### 2.6. Quantification of CYP2E1 and PEPCK Protein Expression

The levels of PEPCK as a PP marker and CYP2E1 as a PV marker were measured by ELISA kit. To form dynamic zonation in the LADY chip, flow was introduced from the PP region to the PV region, whereas no flow was formed in the static chip used as a control. After 3 days of culture, reversibly bonded microfluidic layers were detached from the glass slide, and the PP and PV cells were collected from zones 1 and 3. Collected cells were lysed using an ultrasonicator to obtain whole protein extracts. Assays were performed according to the manufacturers’ instructions by using a human CYP2E1 ELISA kit (LifeSpan Biosciences, Inc., Seattle, WA, USA) and a PEPCK assay kit (Abcam, ab239714). The absorbance was measured as in Section 2.5.

### 2.7. RT-PCR

Zone-specific gene expression was analyzed using 2-step real-time PCR (RT-PCR). Following 3-day culture in the presence of flow, reversibly bonded microfluidic layers were detached from the glass slide, and the cells were recovered from zones 1 and 3. RNA was extracted using an AccuPrep^®^ Universal RNA Extraction kit according to the manufacturer’s instructions. Then, AccuPower^®^ RocketScriptPowertracti was used to synthesize cDNA from total RNA as a template according to the user’s guide with an oligo (dT) 20 mer. AccuPower^®^ 2X GreenStar™ qPCR Master Mix was used for RT-PCR to amplify and detect cDNA targets following the vendor’ protocol. RT-PCR was performed in a StepOnePlus™ Real-Time PCR System (Applied Biosystems^®^ by Life technologies™, Waltham, CA, USA). The following primer pairs for human genes were used: *pck2* (forward 5′-gggtgctagactggatctgc-3′, reverse 5′-ctggttgacctgctctgtca-3′; reference NM_004563.4) and *cyp2e1* (forward 5′-acccgagacaccattttcag-3′, reverse 5′-tccagcacacactcgttttc-3′; reference NM_000773.4). The thermal cycling conditions were as follows: 95 °C for 10 min, 40 cycles at 95 °C for 15 s and 60 °C for 1 min, 95 °C for 15 s, 60 °C for 1 min, and 95 °C for 15 s. The mRNA level of each target gene was analyzed in triplicate by the comparative Ct (ΔΔCt) method and normalized to that of GAPDH. All kits and the oligo (dT) 20 mer were purchased from Bioneer (Daejeon, Korea).

### 2.8. Drug-Induced Hepatotoxicity Test

Cell viability was tested with propidium iodide (PI, Sigma-Aldrich). After the cells were cultured for 48 h in the LADY chip, they were treated with 0 (control), 5, 10, or 20 mM acetaminophen (APAP). Following 24 h APAP exposure, 2.5 µM PI was added to label the nuclei of damaged cells and compare viability between PP cells in zone 1 and PV cells in zone 3. Fluorescent images were taken by IXM-C, and cell viability was quantified by using ImageJ software (NIH, Bethesda, MD, USA).

### 2.9. Statistical Analysis

All quantitative data are presented as the mean ± standard error of the mean (SEM) from three devices (n = 3). Protein and mRNA levels, cell viability, and staining intensity were assessed using unpaired Student’s *t*-test and one-way ANOVA followed by a Tukey’s test. To perform statistic calculations, the GraphPad Prism 8 software (San Diego, CA, USA) was used. *p*-values less than 0.05 (95%) were considered to indicate statistical significance.

## 3. Results

### 3.1. Optimal Conditions for Cell Culture in the LADY Chip

Optimal conditions for hepatocyte (HepG2) cultures in the LADY chip were determined to accurately test drug-induced toxicity. HUVECs were injected into the PP well, whereas HepG2 cells were injected into the PV well to obtain a combination of sinusoidal cells-like endothelial cells and adjacent hepatocyte tissue at the boundary of the weir structure. HepG2 cells were trapped by a 7 μm gap of the weir and arranged with HUVECs in a partial radial pattern, which corresponds to half of the diamond shape of the liver acinus. The hydrophilicity of the inner surface of the LADY chip induced by plasma treatment enabled HepG2 cells to be quickly stacked in the cell chamber. The resultant LADY chip was incubated for 1 h at 37 °C to stabilize the stacked cells, although longer stabilization time increases cell viability. The reason for the 1 h incubation was that some bubbles formed inside the chamber at longer incubation times, enhancing the proportion of dead cells.

We determined the optimal volume of culture medium in the PP and PV wells to maintain long-term cell viability (Figure 2a). After 1 h cell stabilization time, culture medium was immediately supplied into both wells to prevent bubble formation and provide fresh nutrients to cells by diffusion. Each well was filled with a total medium volume of 50 μL in five devices, with a different pattern of successive filling in each device: (i) 5 × 10 μL; (ii) 20 μL, 3 × 10 μL; (iii) 30 μL and 2 × 10 μL; (iv) 40 μL and 10 μL; and (v) 50 μL. Testing cell viability by staining dead cells with PI showed that filling both wells with 10 μL of medium 5 times was the best, since first adding a lower volume of culture medium causes less shear stress to the cells, leading to high cell viability. Shear stress is known to affect hepatocyte viability and functionality [35].

To determine the optimal volume of culture medium for delivery of oxygen and nutrients to cells, after 3 h incubation we induced a gradual flow of culture medium from the PP to the PV well by adding different volumes of medium into each well. Culture medium flow is driven by gravity caused by the height difference of the medium between the PP and PV wells: the greater the difference, the faster the flow. Each PP well was filled with 300 μL of culture medium, whereas the volume in the PV well varied and was 200, 150, or 100 μL. Filling the PV well with 150 μL of culture medium resulted in highest cell viability (Figure 2b). Lower viability with 100 μL of medium in the PV well might be caused by an increased shear stress because flow was too fast. On the other hand, the lowest viability with 200 μL of medium may be due to the insufficient delivery of oxygen and nutrients to cells.

### 3.2. Liver Acinus–Mimetic Cell Pattern and Functionality of Hepatocytes in the LADY Chip 

As a proof of concept, the liver acinus–mimetic cell pattern in the LADY chip was demonstrated with lentiviral transduced HepG2 and HUVECs (Figure 3). Highly purified FACS-sorted HepG2 cells producing GFP (95% purity) and HUVECs expressing RFP (93% purity) were co-cultured in the device for 12 days, and their confocal images showed that HepG2 cells and HUVECs were alternately arranged, demonstrating a successfully established liver acinus–mimetic pattern. Their fluorescence intensities were stable for 12 days (data not shown), thus verifying high cell viability in long-term culture under optimized experimental conditions in the LADY chip.

To assess liver function and investigate the effect of co-culture of HepG2 cells with HUVECs in comparison with HepG2 mono-culture in the device, we also measured the levels of the liver-specific biomarkers albumin and urea. Generally, measurement of albumin secretion is used as a liver function test to indicate potential liver injury, and urea secretion is also an indicator of essential liver metabolism and xenobiotic function through ammonia metabolism. Quantification of the production of albumin and urea on days 2–7 of HepG2 mono-culture and HepG2/HUVEC co-culture was determined (Figure 4). The levels of albumin secretion in both devices peaked with the maximum metabolic activity on day 3 of culture and then slightly decreased (Figure 4a). This can be explained by higher proliferation and functionality in the beginning of the culture, which led to higher metabolic activity [36]. Average albumin production during the culture period was 167 ± 4 ng/h/10^6^ cells in the mono-culture and 172 ± 4 ng/h/10^6^ cells in the co-culture. These values are similar or higher to the result (125 ± 41 ng/h/10^6^ cells) for the previously reported HepG2-based microfluidic device [37]. Urea production levels were high, above 170 nM/h/10^6^ cells, during the whole culture period in both devices (Figure 4b), and these results are also in line with previous reports of a urea production level around 145 nM/h/10^6^ of HepG2 cells cultured in a microfluidic chip [38].

The levels of albumin secretion and urea production showed higher trend in HepG2/HUVECs co-culture than in HepG2 mono-culture over the whole culture period (Figure 4). This difference could be caused by the dependence of these albumin secretion and urea production activities on interactions between parenchymal liver cells and non- parenchymal cells, which regulates hepatic functions [39,40]. These findings emphasize that this co-culture device with HepG2 cells and HUVECs could have better potential for testing of drug-induced toxicity than mono-culture devices.

### 3.3. Hepatic Zonation in the LADY Chip

We demonstrated zonal heterogeneity to prove the formation of hepatic zonation and confirm the possibility of screening for drug-induced zonal hepatotoxicity using our LADY chip. To collect zonated cells, we detached the upper microfluidic PDMS of the device from the glass slide after oxygen/glucose-carrying medium had dynamically and continuously flowed from the PP well to the PV well via the narrow weir gap over 3 days of culture. As a control, we also prepared a static chip without flow. Then, we carefully collected hepatic cells located in PP zone 1 and PV zone 3, and quantified their relative protein levels of PEPCK as a PP marker and CYP2E1 as a PV marker and the mRNA levels of *pck2* (PEPCK gene) and *cyp2e1* (CYP2E1 gene). 

Interestingly, the PEPCK protein level in zone 1 cells was 2.5 times higher than in zone 3 cells in the dynamic chip, whereas there was no noticeable difference in the static chip (Figure 5a). On the contrary, the CYP2E1 protein level in zone 3 was about 3 times higher than in zone 1, but the static chip showed no noticeable difference between the two zones (Figure 5b). Consistent with the protein levels, the *pck2* mRNA expression level was significantly higher in zone 1 than in zone 3 (Figure 5c), whereas that of *cyp2e1* was slightly higher in zone 3 than in zone 1 (Figure 5d). These results support the view that PEPCK, a key enzyme in gluconeogenesis, is predominantly found in the PP region (zone 1), where oxygen-rich blood from the hepatic artery blends with nutrient-rich blood from the portal vein and that CYP enzymes are widely expressed and commonly induced by chemicals in the PV region (zone 3). Cells in zone 1 faced medium flow first and consumed the dissolved oxygen delivered by medium. Hence, cells in zone 3 experienced a lower amount of oxygen. The predominant expression of CYP2E1 in zone 3 is strongly related to the oxygen gradient and Wnt/β-catenin pathway, which is known to be a key signaling pathway regulating the expression of drug-metabolizing enzymes such as CYPs in the liver [41,42]. The Wnt/β-catenin pathway can be modulated by the hypoxia signaling system and hypoxia-inducible transcription factors [27,43]. It is assumed that zonated protein expression observed among cells of the same type in different locations is caused by gradients of regulators such as nutrients, oxygen, cytokines, and other signaling molecules within the liver acinus, yielding different rates of transcription, mRNA translation and degradation, or protein degradation along the hepatic sinusoids [44]. Furthermore, the dynamic flow of these zonation factors such as oxygen and nutrients from zone 1 to zone 3 in the device might lead to the zonated expression pattern, which resulted in the remarkable differences of PEPCK and CYP2E1 levels between the dynamic and static devices. These findings indicate that our LADY chip successfully recapitulated the microenvironment and metabolic zonation of the liver acinus.

### 3.4. Drug-Induced Zonal Hepatotoxicity

To test drug-induced zonal hepatotoxicity and the effect of non-parenchymal cells on hepatotoxicity in our LADY chip, we cultured HepG2 cells with or without HUVECs and used APAP, which is known to cause severe hepatic necrosis, as a test drug. A medium flow formed by a volume difference between the PP and PV wells induced hepatic zonation from zone 1 toward zone 3. The oxygen/glucose-containing medium was first delivered to HUVECs and then encountered HepG2 cells in zone 1, 2, and 3, consecutively, by passing through the gap in the weir structure. After continuous medium flow was generated for 48 h, HepG2 cells were exposed to various concentrations (0, 5, 10, or 20 mM) of APAP with flow for another 24 h at 37 °C. After dosing with APAP, the cells were stained with PI to assess viability. Confocal microscopy was then used to image dead cells within zones 1 and 3.

A gradient of cell viability was observed across the length of the culture chamber due to the formation of metabolic zonation (Figure 6). The PI images indicated that HepG2 cells were injured in an APAP dose–dependent manner in both the mono-culture of HepG2 cells and their co-culture with HUVECs (Figure 6a). As expected, HepG2 cells were more damaged in zone 3 than in zone 1 in both devices, although the APAP-containing medium came into contact with the cells in zone 1 first. More HepG2 cell damage in zone 3 was caused by a lower oxygen content in zone 3 than in zone 1. A difference in CYP expression between zones 1 and 3 in the LADY chip containing HepG2 cells was formed according to a gradual change in oxygen concentration: the low oxygen content in zone 3 yielded the high expression of CYP, which is a representative metabolic enzyme converting APAP to N-acetyl-p-benzoquinoneimine, producing superoxide anion and depleting GSH [45]. The superoxide anion reacts with nitric oxide, resulting in formation of peroxynitrite, which is highly reactive and causes mitochondrial damage, whereas GSH functions as a scavenger of reactive oxygen species and peroxynitrite. The elevated CYP activity in zone 3 activated APAP metabolism, resulting in a greater formation of reactive oxygen and nitrogen species in zone 3 than in zone 1. This increased oxidative stress caused the loss of mitochondrial membrane potential, leading to cell necrosis in zone 3 [46,47].

HepG2 cell viability was enhanced by co-culturing them with HUVECs in both zones 1 and 3 at each APAP concentration (Figure 6b). Recent studies have demonstrated that co-culture of hepatocytes with endothelial cells not only improves hepatic functions but also enhances resistance to APAP cytotoxicity in comparison with the hepatocyte mono-culture model [48,49]. Co-culture increases the levels of nitric oxide, which activates antioxidant transcription factors, increasing synthesis of intracellular GSH as a scavenger. The increased GSH could have a hepatoprotective effect from oxidative stress caused by APAP toxicity. Overall, these results show that our device is a useful physiologically relevant in vitro platform for assessing aspects of APAP-induced zonal hepatotoxicity.

## 4. Conclusions

Liver zonation is particularly important in understanding liver function and drug-induced zonal hepatotoxicity. In this study, we have developed a liver acinus–mimetic zonation chip that successfully recapitulated the microstructure of the liver acinus and hepatic zonation in vitro. A part of the hexagonal shape of the liver lobule was formed using a radial pattern with alternating HepG2 cells and HUVECs, and continuous medium flow was created in the same direction as blood flows from portal triads to the central vein. To form gradual changes in metabolic functions of liver cells in different zones, some aspects of metabolic zonation were created by the flow of oxygen/glucose-containing medium to generate gradients of zonation factors in the device. We confirmed that periportal and perivenous markers were expressed in the cells located in zone 1 and 3, respectively, and zonal necrosis in zone 3 was caused by a hepatotoxic drug. Accordingly, our device could provide information more relevant to toxicities in humans than that obtained using traditional 2D cell culture or nonclinical animal models. Thus, recapitulation of the key architecture and specific functional features of the liver acinus in our device provides a human-relevant physiological environment, thereby enabling cellular responses similar enough to in vivo responses. This platform also provides a simple chip fabrication step and cell injection method as well as a pumpless system that can be used as a high-throughput tool for the testing of candidate hepatotoxic substances and chemical risk assessment at the human systemic level. Finally, our new device can be re-purposed to study liver diseases, patient-specific developmental responses for personalized medicine, and targeted differentiation of induced pluripotent stem cells to mature hepatocytes. We plan to use primary human hepatocytes and liver sinusoidal endothelial cells as well as Kupffer and hepatic stellate cells as additional non-parenchymal cells in our further research.

## Figures and Tables

**Figure 1 biosensors-12-00445-f001:**
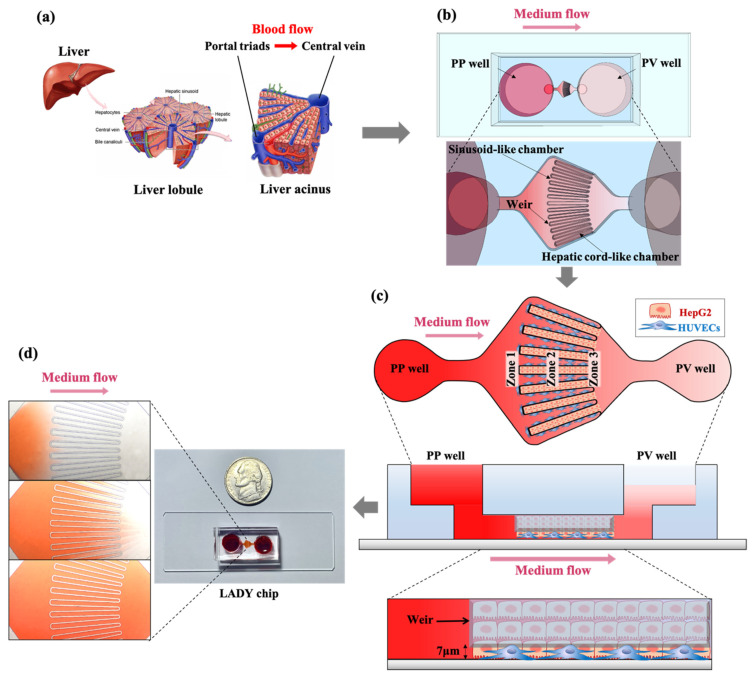
Schematic of the liver acinus dynamic (LADY) chip. (**a**) The design of the LADY chip was inspired by the microenvironment of the liver acinus in the lobule. (**b**) shows a top view and microstructure of the LADY chip. Hepatic cord–like cell chambers are alternately arranged with sinusoid-like chambers in a radial shape. (**c**) details co-culture of HepG2 and HUVECs in the LADY chip. HUVECs injected into the PP well and HepG2 cells applied into the PV well were trapped by a weir structure which is placed 7 µm above the bottom and arranged in a partial radial pattern, which recapitulates the microstructure of the liver acinus. On the basis of the functional characteristics of the liver acinus and the direction of blood flow, metabolic zonation from zone 1 to 3 was formed by medium flow from the PP well to the PV well, passing through the fenestrated weir. A photograph of the LADY chip with red dye-filled microchambers is shown in (**d**).

**Figure 2 biosensors-12-00445-f002:**
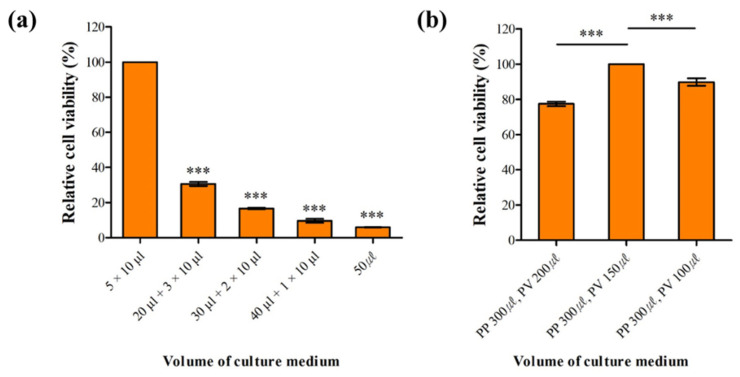
Optimization of culture conditions to improve cell viability in the LADY chip. (**a**) Following stabilization and attachment of HepG2 cells after injection into the hepatocyte chamber, additional culture medium was supplied into the periportal (PP) and the perivenous (PV) well to provide fresh nutrients and prevent bubble formation. The medium (50 μL) was added in various patterns as indicated in the horizontal axis labels. (**b**) After cell stabilization in the cell chamber, culture medium gradually flowed from the PP well to the PV well to produce gradients of zonation factors for the formation of metabolic zonation in the LADY chip. Medium flow is driven by gravity and is caused by height difference between two reservoirs (the greater the height difference, the faster the flow). To vary the flow rate, different volumes of culture medium were added to the PP and PV wells as indicated in the horizontal axis labels. Cell viability was tested by staining dead cells with PI under each experimental condition. Error bars show mean ± SEM. *** *p* < 0.001; each experiment was replicated using three different chips (One-way ANOVA, Tukey’s test).

**Figure 3 biosensors-12-00445-f003:**
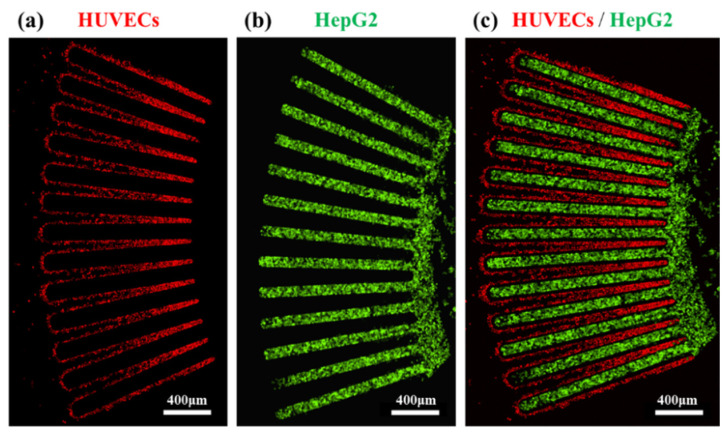
Morphological demonstration of the liver acinus–like radial pattern of HepG2 cells and HUVECs. To monitor heterogeneous cells in long-term culture, HepG2 cells and HUVECs were transduced with GFP and RFP lentiviral vector, respectively. Confocal images of (**a**) RFP-expressing HUVECs, (**b**) GFP-expressing HepG2 cells, and (**c**) co-cultured RFP-expressing HUVECs and GFP-expressing HepG2 cells were taken after they were injected into each cell chamber of the LADY chip. The fluorescence intensities of transduced cells were stable in a radial pattern for 12 days. Scale bars 400 μm.

**Figure 4 biosensors-12-00445-f004:**
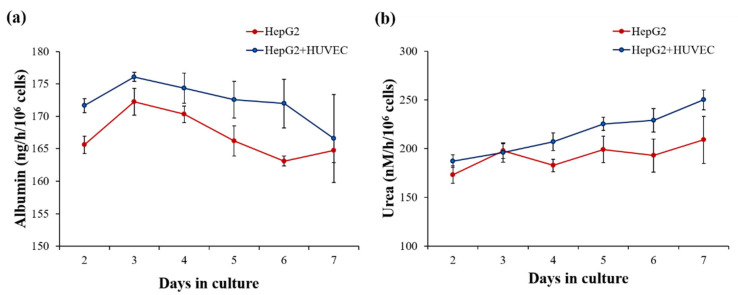
Functional tests of production of liver-specific markers by HepG2 cells cultured in the LADY chip with or without HUVECs. (**a**) Albumin secretion and (**b**) urea synthesis were measured on culture day 7 and compared. Albumin secretion peaked around 175 ng/h/10^6^ cells on day 3 both with and without HUVECs. Urea synthesis levels were maintained over 173 nM/h/10^6^ cells throughout the culture period. Error bars show mean ± SEM. Statistical significance of differences was determined by two-tailed unpaired Student’s *t*-test, and *p* > 0.05 in all data points.

**Figure 5 biosensors-12-00445-f005:**
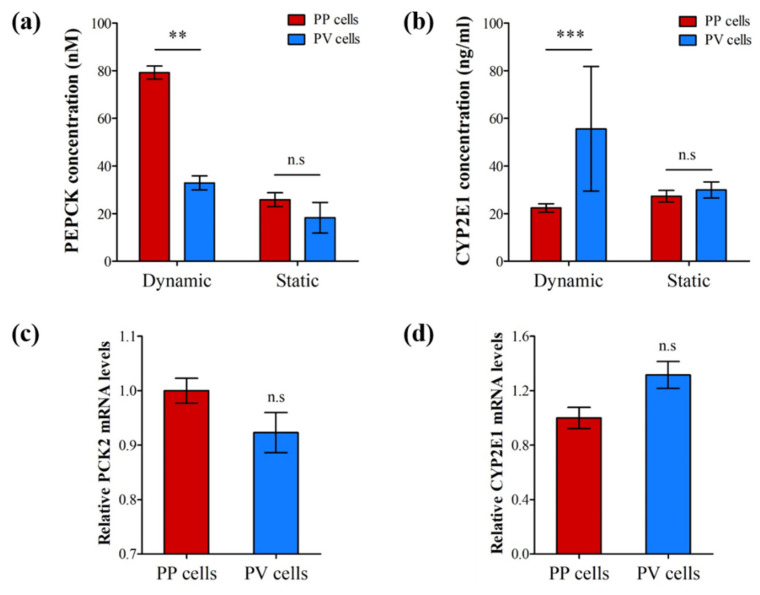
Zonated expression of a periportal (PP) and a perivenous (PV) marker in PP and PV HepG2 cells under dynamic culture condition in comparison with the static control. Protein levels of (**a**) PEPCK as a PP marker and (**b**) CYP2E1 as a PV marker were quantified by ELISA after oxygen/glucose gradients were generated for 3 days in the LADY chips. No fluid flow was formed from the PP to PV well in the static controls. Relative mRNA levels of (**c**) PEPCK gene (*pck2*) and (**d**) CYP2E1 gene (*cyp2e1*) were measured using RT-qPCR after the PP and PV HepG2 cells were recovered from zones 1 and 3, respectively. Error bars show mean ± SEM. Statistical significance of differences was determined by two-tailed unpaired Student’s *t*-test, n.s., not significant (*p* > 0.05), ** *p* < 0.01, *** *p* < 0.001.

**Figure 6 biosensors-12-00445-f006:**
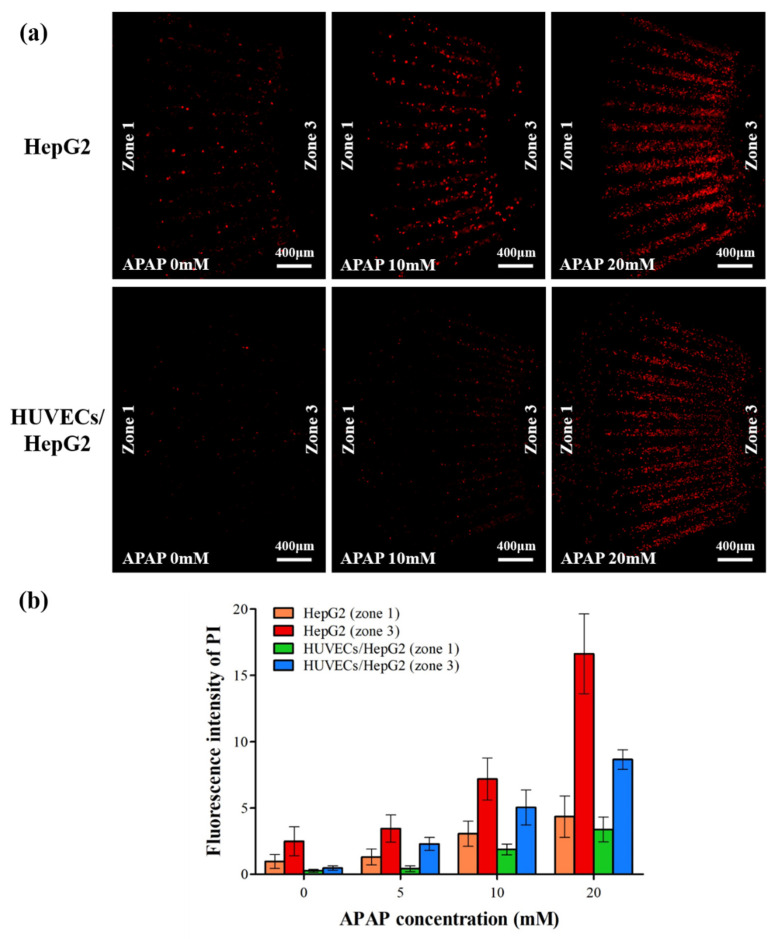
Zonal hepatotoxicity after acetaminophen (APAP) treatment in HepG2 mono-culture model and HepG2/HUVECs co-culture. (**a**) After gradual medium flow was created for 3 days to induce metabolic zonation, cells were exposed to 0, 5, 10, or 20 mM APAP for 24 h. Cells were stained with PI, and confocal images were taken. (**b**) Relative viability of periportal cells located in zone 1 and perivenous cells located in zone 3. Fluorescence of PI was measured and analyzed by ImageJ. Error bars show mean ± SEM. Statistical significance of differences was determined by one-way ANOVA and Tukey’s test. Representative PP and PV zones from 5 individual LADY chips were used for quantification. Scale bars 400 μm.

## Data Availability

Not applicable.

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
