# Peer review of "Liver Acinus Dynamic Chip for Assessment of Drug-Induced Zonal Hepatotoxicity"

_biosensors, 2022, doi:10.3390/bios12070445_

Round 1
Reviewer 1 Report
Thank authors for presenting their work. The research problem seems to be important and interesting. However, a lot of Liver-on-Chips models were designed and produced so far. Thant is why my questions an advice to authors:
1. There is lack of novelty in chip geometry. I found several examples in literature which showed really similar design of the chip. Can you show, explain why your chip is better than published so far?
2. Why you used the DPBS to increase the hydrophilicity of microfluidic chamber. How it looks form chemistry side? DPBS is only standard buffer.
3. Why HUVECs were cultured in DMEM media. They need special supplements. They should be cultured in EGM media.
4. Please provide information about passage number of both cell lines. This is important information, because HUVEC are active only till 8-9 passage.
5. Explain ratio of HepG2:Huvecs cells. Why this one?
6. HUVECs are not good cells for formation of liver tissue. This tissue has specific sinusoidal cells. HUVEC are too general.
7. Liver tissue has much more type of cells. Explain why you choose tony this one, because a ot of researcher used at least 3.
8. HepG2 are cancer cells. Do you think it has effect on good reproduction of liver tissue in vitro activity?
9. Please underline the novelty of research (methods, analysis).
Author Response
Response to Reviewer 1 Comments
Point 1: There is lack of novelty in chip geometry. I found several examples in literature which showed really similar design of the chip. Can you show, explain why your chip is better than published so far?
Response 1: Even though there are several studies that mimic microenvironment of the liver lobule, the novelty of our device is the first model to recapitulate liver zonation in microphysiological structure of the liver acinus so far. That the zonation phenomenon in the liver acinus structure was implemented in our chip means that it successfully mimicked the important structure of the human liver and is useful to improve the human relevance of toxicity testing results. Besides, the unique weir structure in the chip could effectively separate HepG2 and HUVEC cells for their co-culture and also supply fresh culture medium and reduce shear stress to them.
Point 2: Why you used the DPBS to increase the hydrophilicity of microfluidic chamber. How it looks form chemistry side? DPBS is only standard buffer.
Response 2: The treatment of oxygen plasma on PDMS introduces polar functional groups which is mainly the silanol group (SiOH), yielding hydrophilicity. We used DPBS without divalent cations, such as calcium and magnesium to keep the hydrophilicity of the microfluidic chamber until use. DBPS was then replaced into cell culture medium, followed by cell injection into the chamber. As a result, any aqueous buffer available for cell culture can be used to keep the hydrophilicity in this chamber.
Point 3: Why HUVECs were cultured in DMEM media. They need special supplements. They should be cultured in EGM media.
Response 3: We used HUVECs (EA.hy926) obtained from ATCC. The manufacture recommended us to use ATCC-formulated DMEM (Catalog No. 30-2002) supplemented with 10% FBS as growth medium for culture of HUVECs.
Point 4: Please provide information about passage number of both cell lines. This is important information, because HUVEC are active only till 8-9 passage.
Response 4: We only used HUVECs and HepG2 at a passage of 3-6 and 50-55, respectively.
Point 5: Explain ratio of HepG2:Huvecs cells. Why this one?
Response 5: A previous study reported that liver is composed of parenchymal cells (80%) and non-parenchymal cells (NPC) (20%), and liver sinusoidal endothelial cells (LSECs) account for 50% among NPC population (Reference 1). Thus, the ratio of hepatocytes to LSECs can be calculated as about 8:1. Based on this ratio, we used a 10:1 ratio of HepG2 to HUVECs in this study with taking error into account.
Reference 1: Pandey, E., Nour, A. S., & Harris, E. N. (2020). Prominent receptors of liver sinusoidal endothelial cells in liver homeostasis and disease. Frontiers in Physiology, 11, 873.
Point 6: HUVECs are not good cells for formation of liver tissue. This tissue has specific sinusoidal cells. HUVEC are too general.
Response 6: We agree with your comments, HUVECs are too general, but the main purpose of our study was to confirm the functionality and potential of our LADY chip developed for drug-induced toxicity testing. Thus, we just used HUVECs the most commonly used models as endothelial cells since they are known to have enough endothelium function including cell type homogeneity, well-characterized surface markers, late endothelial progenitor cell characters, and obtainable huge cell numbers via rapid expansion property (Reference 2). Moreover, the representative function of liver sinusoidal endothelial cells is a permeable barrier that lack of basement membrane and fenestrations permit relatively free movement of macromolecules between the blood and the liver parenchyma. The unique weir structure of our device could act as a fenestrated endothelium which supplies fresh culture medium and reduces shear stress to cells, which could contribute to overcome somewhat the drawbacks of HUVECs. Also, we plan to continue research on the safety and efficacy of drugs in patients with liver diseases via the LADY chip developed here, so we will use specific sinusoidal cells in further study, as you suggested. Thanks for your valuable comments.
Reference 2: Wu, C. C., Chen, Y. C., Chang, Y. C., Wang, L. W., Lin, Y. C., Chiang, Y. L., ... & Huang, C. C. (2013). Human umbilical vein endothelial cells protect against hypoxic-ischemic damage in neonatal brain via stromal cell-derived factor 1/CXC chemokine receptor type 4. Stroke, 44(5), 1402-1409.
Point 7: Liver tissue has much more type of cells. Explain why you choose tony this one, because a ot of researcher used at least 3.
Response 7: We agree with you. It is ideal to co-culture as many types of cells as possible. However, in this study, we preferentially focused on our main purpose to recapitulate the structure of the liver acinus important in liver tissue, the formation of liver zonation in the structure, and the effect of endothelial cells on drug-induced hepatotoxicity. As you suggested, we will plan to add Kupffer cells and hepatic stellate cells besides hepatocytes and liver sinusoidal cells in further research.
Point 8: HepG2 are cancer cells. Do you think it has effect on good reproduction of liver tissue in vitro activity?
Response 8: We also agree with your comments. Expression levels of enzymes in HepG2 cells differ significantly from that in human as they are cancer cell line. However, in this study we used HepG2 cells because they are cost effect, easy to use, and provide an unlimited supply. As mentioned above, we tried to confirm the functionality of our new device mimicking the structure of liver acinus for drug-induced toxicity testing in this study. As you pointed out, we will plan to use primary human hepatocytes in further study. Thanks much again.
Point 9: Please underline the novelty of research (methods, analysis).
Response 9: As mentioned above, the novelty of our device is that it recapitulated microphysiological structure of the liver acinus as well as liver zonation in vitro. Furthermore, the device provides simple chip fabrication method since it doesn’t need ports to connect tubing with a syringe pump. Also, it allows us to easily inject cells into the device only with a pipette and create a continuous medium flow by gravity without any external pump. It is not only easy-to-use, cost-effective, reproducible, but also enables high-throughput screening of a lot of drug candidates. Besides, it is compatible with standard bio-lab equipment and assays such as fluorescent microscopy, cell viability test, and quantification of protein and gene expression by harvesting of cells cultured in the device.
Reviewer 2 Report
This paper is clearly written and in a logical order. It covers all the standard steps in a typical liver co-culture on a chip. I only have a few problems.
Please explain why you haven't used a continuous perfusion culture with a pump?
Does the incremental addition of the media at the PP well have any effect on the cells? It would be a start-stop (or nearly stop) flow, with decreasing flow rate. Wouldn't Fig.2 indicate that a continuous flow would be best, as making the pipette volumes smaller results in better cell viability?
How do you create a flow anyway, if you add medium to bot PP and PV well? Figure 2 states that media was added to the PP and PV well.
How was the waste removed from the chip? And wouldn't you have to remove the waste and supply the fresh medium at the same time?
The summary at the end of section 3 sounds like a conclusion and is more or less a duplicate of the conclusion.
more specific points.
line 20: "that" is in the wrong place
line 287: delete "was"
line 315: I do not think that a 5 ng difference on albumin production is a significant difference between mono and co-culture, especially since the erro is +/- 4 ng.
Figure 4: How come a number of data points do not have error bars? Were these measurements only performed once? If so, please do some more experiments to have a statistical relevant result. A single measurement means nothing.
Author Response
Response to Reviewer 2 Comments
Point 1: Please explain why you haven't used a continuous perfusion culture with a pump?
Response 1: Perfusion system with a syringe pump needs an external device, which is highly cost and makes it hard to perform high-throughput screening. Therefore, we used a gravity-driven flow system to make hydrostatic pressure difference between inlets and outlets, resulting in continuous fluid flow through the channels without the need for external pumps and tubing. Also, this gravity-driven flow system is easy, cost-effective to construct, and useful for high-throughput screening.
Point 2: Does the incremental addition of the media at the PP well have any effect on the cells? It would be a start-stop (or nearly stop) flow, with decreasing flow rate. Wouldn't Fig.2 indicate that a continuous flow would be best, as making the pipette volumes smaller results in better cell viability?
Response 2: Cells are exposed to shear stress under the continuous fluid flow in the perfusion system. Although a small volume (50 μl) of medium was added into both the PP and PV wells, the flow in a microchannel could be driven and maintained over a period of time by capillary force in the device. As shown in Fig. 2, adding wells with 10 μl of medium 5 times could reduce shear stress to cells whereas the addition of 50 μl of medium at a time caused more shear stress to them, resulting in low cell viability because the cells were very unstable at the beginning of culture in the device.
Point 3: How do you create a flow anyway, if you add medium to both PP and PV well? Figure 2 states that media was added to the PP and PV well.
Response 3: At the very beginning of cell culture, a flow was created in microfluidic channels after a small volume of medium was added into both the PP and PV wells by capillary force. After cell stabilization, we created a flow of culture medium from the PP to the PV well by adding different volumes of medium into each well. Culture medium flow is driven by gravity caused by the height difference of the medium between the PP and PV wells: the greater the difference, the faster the flow.
Point 4: How was the waste removed from the chip? And wouldn't you have to remove the waste and supply the fresh medium at the same time?
Response 4: PP well acts as a medium inlet whereas PV well acts as medium outlet. Higher volume and lower volume of culture medium were added to the PP and PV wells, respectively, which causes a medium flow from the PP to PV well by gravity. During cell culture, wastes were flowed out through the PV well by supplying fresh medium into the PP well at the same time.
Point 5: The summary at the end of section 3 sounds like a conclusion and is more or less a duplicate of the conclusion.
Response 5: Changed as suggested. We deleted the summary at the end of section 3. Thanks for your valuable comments.
Point 6: line 20: "that" is in the wrong place
Response 6: Corrected as suggested. Thanks much.
Point 7: line 287: delete "was"
Response 7: Corrected as suggested. Thanks much.
Point 8: line 315: I do not think that a 5 ng difference on albumin production is a significant difference between mono and co-culture, especially since the error is +/- 4 ng.
Response 8: We agree with you, so we changed the sentence in line 321-323, “The levels of albumin secretion and urea production were significantly higher in HepG2/HUVECs co-culture than in HepG2 mono-culture over the whole culture period (Figure 4)” to “The levels of albumin secretion and urea production showed a higher trend in HepG2/HUVECs co-culture than in HepG2 mono-culture over the whole culture period (Figure 4)”.
Point 9: Figure 4: How come a number of data points do not have error bars? Were these measurements only performed once? If so, please do some more experiments to have a statistical relevant result. A single measurement means nothing.
Response 9: We corrected “Figure 4” as you pointed out. Thank you for your comments.
Reviewer 3 Report
Out of journal scope.
Author Response
Thank you for your review.
Round 2
Reviewer 1 Report
Thanks for the answers. I am satisfied. However, please include the answers in the main manuscript. For example, information on passages number, information about novelty, why this ratio od HepG2:HUVEC, why HUVEC and not sinusoidal endothelial cells, etc.
Author Response
Point 1: There is lack of novelty in chip geometry. I found several examples in literature which showed really similar design of the chip. Can you show, explain why your chip is better than published so far?
Response 1: Even though there are several studies that mimic microenvironment of the liver lobule, the novelty of our device is the first model to recapitulate liver zonation in microphysiological structure of the liver acinus so far. That the zonation phenomenon in the liver acinus structure was implemented in our chip means that it successfully mimicked the important structure of the human liver and is useful to improve the human relevance of toxicity testing results. Besides, the unique weir structure in the chip could effectively separate HepG2 and HUVEC cells for their co-culture and also supply fresh culture medium and reduce shear stress to them.
>> As you suggeted, we added the novelty sentence to the end of introduction part. Thank you.
Point 2: Why you used the DPBS to increase the hydrophilicity of microfluidic chamber. How it looks form chemistry side? DPBS is only standard buffer.
Response 2: The treatment of oxygen plasma on PDMS introduces polar functional groups which is mainly the silanol group (SiOH), yielding hydrophilicity. We used DPBS without divalent cations, such as calcium and magnesium to keep the hydrophilicity of the microfluidic chamber until use. DBPS was then replaced into cell culture medium, followed by cell injection into the chamber. As a result, any aqueous buffer available for cell culture can be used to keep the hydrophilicity in this chamber.
>> Accordingly, page 4, line 136-137, at the end of section 2.1. “both the PP well……… to increase……cell chamber” changed to “both the PP well……… to maintain…….into the chamber”.
Point 3: Why HUVECs were cultured in DMEM media. They need special supplements. They should be cultured in EGM media.
Response 3: We used HUVECs (EA.hy926) obtained from ATCC. The manufacture recommended us to use ATCC-formulated DMEM (Catalog No. 30-2002) supplemented with 10% FBS as growth medium for culture of HUVECs.
Point 4: Please provide information about passage number of both cell lines. This is important information, because HUVEC are active only till 8-9 passage.
Response 4: We only used HUVECs and HepG2 at a passage of 3-6 and 50-55, respectively.
>> As you suggested, we added as following a sentence: page 4, line 155, “HepG2 cells and HUVECs were used at a passage of 3-6 and 50-55, respectively.”
Point 5: Explain ratio of HepG2:Huvecs cells. Why this one?
Response 5: A previous study reported that liver is composed of parenchymal cells (80%) and non-parenchymal cells (NPC) (20%), and liver sinusoidal endothelial cells (LSECs) account for 50% among NPC population [34]. Thus, the ratio of hepatocytes to LSECs can be calculated as about 8:1. Based on this ratio, we used a 10:1 ratio of HepG2 to HUVECs in this study with taking error into account.
>> As you suggested, we added the above sentence to the end of section 2.3. References were also added into revised manuscript.
Reference 34: Pandey, E., Nour, A. S., & Harris, E. N. (2020). Prominent receptors of liver sinusoidal endothelial cells in liver homeostasis and disease. Frontiers in Physiology, 11, 873.
Point 6: HUVECs are not good cells for formation of liver tissue. This tissue has specific sinusoidal cells. HUVEC are too general.
Response 6: We agree with your comments, HUVECs are too general, but the main purpose of our study was to confirm the functionality and potential of our LADY chip developed for drug-induced toxicity testing. Thus, we just used HUVECs the most commonly used models as endothelial cells since they are known to have enough endothelium function including cell type homogeneity, well-characterized surface markers, late endothelial progenitor cell characters, and obtainable huge cell numbers via rapid expansion property (Reference 33). Moreover, the representative function of liver sinusoidal endothelial cells is a permeable barrier that lack of basement membrane and fenestrations permit relatively free movement of macromolecules between the blood and the liver parenchyma. The unique weir structure of our device could act as a fenestrated endothelium which supplies fresh culture medium and reduces shear stress to cells, which could contribute to overcome somewhat the drawbacks of HUVECs. Also, we plan to continue research on the safety and efficacy of drugs in patients with liver diseases via the LADY chip developed here, so we will use specific sinusoidal cells in further study, as you suggested. Thanks for your valuable comments.
>> References were added into the revised manuscript.
Reference 33: Wu, C. C., Chen, Y. C., Chang, Y. C., Wang, L. W., Lin, Y. C., Chiang, Y. L., ... & Huang, C. C. (2013). Human umbilical vein endothelial cells protect against hypoxic-ischemic damage in neonatal brain via stromal cell-derived factor 1/CXC chemokine receptor type 4. Stroke, 44(5), 1402-1409.
Point 7: Liver tissue has much more type of cells. Explain why you choose tony this one, because a ot of researcher used at least 3.
Response 7: We agree with you. It is ideal to co-culture as many types of cells as possible. However, in this study, we preferentially focused on our main purpose to recapitulate the structure of the liver acinus important in liver tissue, the formation of liver zonation in the structure, and the effect of endothelial cells on drug-induced hepatotoxicity. As you suggested, we will plan to add Kupffer cells and hepatic stellate cells besides hepatocytes and liver sinusoidal cells in further research.
Point 8: HepG2 are cancer cells. Do you think it has effect on good reproduction of liver tissue in vitro activity?
Response 8: We also agree with your comments. Expression levels of enzymes in HepG2 cells differ significantly from that in human as they are cancer cell line. However, in this study we used HepG2 cells because they are cost effect, easy to use, and provide an unlimited supply. As mentioned above, we tried to confirm the functionality of our new device mimicking the structure of liver acinus for drug-induced toxicity testing in this study. As you pointed out, we will plan to use primary human hepatocytes in further study. Thanks much again.
Point 9: Please underline the novelty of research (methods, analysis).
Response 9: As mentioned above, the novelty of our device is that it recapitulated microphysiological structure of the liver acinus as well as liver zonation in vitro. Furthermore, the device provides simple chip fabrication method since it doesn’t need ports to connect tubing with a syringe pump. Also, it allows us to easily inject cells into the device only with a pipette and create a continuous medium flow by gravity without any external pump. It is not only easy-to-use, cost-effective, reproducible, but also enables high-throughput screening of a lot of drug candidates. Besides, it is compatible with standard bio-lab equipment and assays such as fluorescent microscopy, cell viability test, and quantification of protein and gene expression by harvesting of cells cultured in the device.
Reviewer 2 Report
All my comments have been addressed. Thanks.
Author Response
Thank you for your valuable comments!